environmental engineering/environmental science

methylene blue, manganese ore, kinetics, dye, decoloration

**Authors for correspondence:**
Yide He
e-mail: heyd@njtech.edu.cn
Yongjun Zhang
e-mail: y.zhang@njtech.edu.cn

This article has been edited by the Royal Society of Chemistry, including the commissioning, peer review process and editorial aspects up to the point of acceptance.

# Degradation of methylene blue by natural manganese oxides: kinetics and transformation products

Shuangxi Zhou, Zhiling Du, Xiuwen Li, Yunhai Zhang, Yide He and Yongjun Zhang

School of Environmental Science and Engineering, Nanjing Tech University, Pu Zhu Nan Lu 30, Nanjing 211800, People's Republic of China

SZ, 0000-0002-9633-3664

In this study, natural manganese oxides ($MnO_x$), an environmental material with high redox potential, were used as a promising low-cost oxidant to degrade the widely used dyestuff methylene blue (MB) in aqueous solution. Although the surface area of $MnO_x$ was only $7.17\,m^2\,g^{-1}$, it performed well in the degradation of MB with a removal percentage of 85.6% at pH 4. It was found that MB was chemically degraded in a low-pH reaction system and the degradation efficiency correlated negatively with the pH value (4–8) and initial concentration of MB ($10$–$50\,mg\,l^{-1}$), but positively with the dosage of $MnO_x$ ($1$–$5\,g\,l^{-1}$). The degradation of MB fitted well with the second-order kinetics. Mathematical models were also built for the correlation of the kinetic constants with the pH value, the initial concentration of MB and the dosage of $MnO_x$. Furthermore, several transformation products of MB were identified with HPLC-MS, which was linked with the bond energy theory to reveal that the degradation was initiated with demethylation.

## 1. Introduction

Nowadays, a great deal of wastewater from industrial production is discharged into the natural environment, leading to a certain degree of contamination [1]. Dyestuff is an essential industrial ingredient, and the manufacturing processes of which release a large volume of wastewater containing dyestuff [2]. It has been estimated that 10–15% of dyestuff in the dyeing process is drained into natural water via sewage plants [3]. Among these dyestuffs, methylene blue (MB) is a traditional dyestuff and is widely applied in papermaking, textile, plastics, cosmetics, etc. [4]. However, some researchers have reported that people

exposed to MB in the environment experience nausea, chest pain, dizziness and headaches, especially infants and pregnant women [5–8]. In addition, the dyestuff wastewater containing MB can weaken intense sunlight and further affect the photosynthetic activity of aquatic life and decrease the aesthetics and diversity of the biological community [9,10].

To control the contamination of dyestuff wastewater, various technologies have been investigated, including physical methods such as micelle-enhanced ultrafiltration [11], nano-filtration [12], adsorption [13–15] and chemical methods; for instance, electrochemical degradation [8], ozone oxidation degradation [16–18], photo-catalytic degradation [4,19,20] and electrocoagulation [21]. However, the high cost and complex operation of these physical [22] and chemical methods [23] significantly hinder their wide application. In addition, dyestuff wastewater is difficult to biodegrade because it has a low B/C ratio ($BOD_5$/$CODcr$ of less than 0.1) [24,25]. Herein, there is an urgent need to find a method with high efficiency, economical feasibility and ease of operation to degrade dyestuff wastewater.

It has been reported that manganese dioxide possesses high oxidation–reduction potential (1.29 V, 25°C) and that synthetic $MnO_2$ has been applied to degrade phenol and aromatic amines [26,27]. Studies have also found that it performs well in treating bisphenol A, bisphenol AF, bisphenol S and cephalosporins [28,29]. In addition, Fei *et al.* [30] found that synthesized $MnO_2$ has a high removal efficiency for Congo red. Cheng *et al.* [31,32] reported that nano-$MnO_2$ could completely destroy HCHO at low temperature. Furthermore, Sekine [33] has improved the performance of nano-$MnO_2$, so that it can be practically used to remove HCHO at room temperature. However, the high cost and complex synthetic steps of synthesized $MnO_2$ have limited its application in wastewater treatment. By contrast, manganese ores consisting of multivalent manganese oxides could be a substitute because of abundant reserves in nature. Recently, studies have demonstrated the good degradation efficiency of organic contaminants by natural manganese oxides ($MnO_x$), such as diclofenac [34], melanin [35] and paracetamolin [36] and emerging organic contaminants [37]. $MnO_x$ was also extensively used in drinking water treatment for the removal of iron and manganese [38]. However, there are too many different and complex compositions of natural manganese ores. Therefore, it is necessary to find a kind of $MnO_x$ with high manganese content and stable effect on removing environmental containments.

In the current study, $MnO_x$ was used as an environmental material to degrade MB. The influence of pH, the dosage of $MnO_x$ and the concentration of MB was studied and the degradation kinetics was established. In addition, a degradation pathway is proposed with the transformation products identified with high-performance liquid chromatography–mass spectrometry (HPLC-MS).

# 2. Experiment

## 2.1. Materials and chemical agents

MB (98.5%), acetic acid (99.8%), sodium acetate (98%), sodium hydrogen phosphate (99%), hydrochloric acid (36–38%) and sodium hydroxide (96%) were used as purchased from Sinopharm Chemical Reagent Co., Ltd (Shanghai). Deionized water was used for preparing all solutions. Natural manganese ore ($MnO_x$, with a relative content of $MnO_2$ 75.38%) was purchased from Qingchong Manganese Co., Ltd, Hunan Province, with a size ranging from 0.075 to 0.12 mm. $MnO_x$ was rinsed with deionized water before tests.

## 2.2. Characterization of $MnO_x$

X-ray diffraction (XRD; Rigaku Smartlab, Japan) equipped with monochromatic high-intensity Cu-K$\alpha$ radiation ($\lambda = 0.154$ nm) was used to analyse $MnO_x$. The operating conditions were as follows: voltage 40 kV, current 40 mA, small-angle range from 0.6° to 5° and wide-angle range from 10° to 80° in a scanning step of $0.02° \, s^{-1}$. Scanning electron microscopy (SEM) images were obtained using a JEOL JSM-7800F (Japan). Fourier transform infrared (FTIR) spectra were obtained with a IRAffinity-1 (Shimadzu) spectrometer. The Brunauer–Emmett–Teller (BET) surface areas were measured with a Micromeritics TriStar II 3020 nitrogen adsorption apparatus. The surface charges zeta point was measured by a zeta-meter (Malvern Nano-ZS90).

## 2.3. Degradation test

Aliquots containing 200 ml of MB solution were added to Erlenmeyer flasks containing $0.1 \, mol \, l^{-1}$ acetate or phosphate as a buffer system. The pH of the solution was adjusted with HCl (0.1 M) or

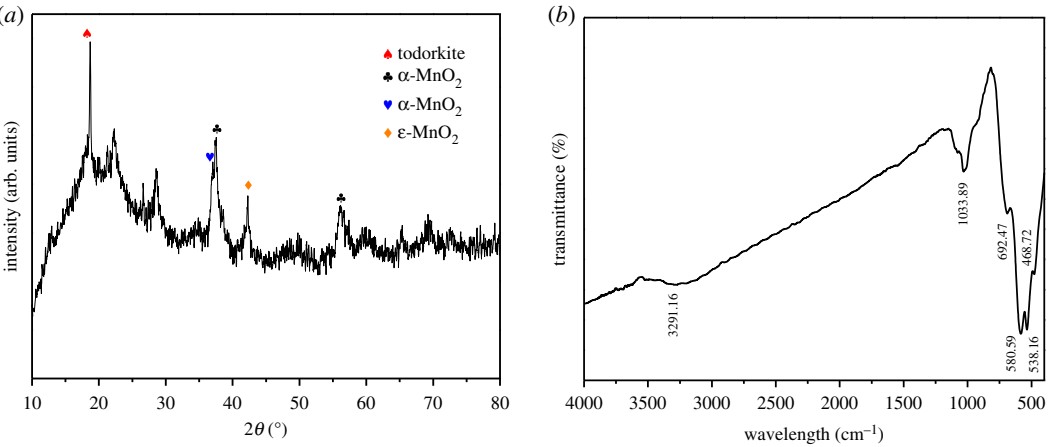

**Figure 1.** Powder XRD patterns of MnO$_x$ (*a*) and infrared spectra of MnO$_x$ (*b*).

NaOH (0.1 M) at room temperature. With stirring on a magnetic stirrer, a certain dosage of MnO$_x$ was added to each flask. A 3.5 ml aqueous sample was collected from the flask at certain time intervals and immediately filtered with a 0.45 μm polyether sulfone (PES) membrane. The filtrate was used to determine the absorbance of the MB solution at a wavelength of 664 nm with a UV–vis spectrophotometer. All experiments were conducted twice and the average value was adopted. The MB degradation percentage was estimated by equation (2.1),

$$\text{degradation percentage of MB (\%)} = \left(1 - \frac{C_t}{C_0}\right) \times 100, \tag{2.1}$$

where $C_0$ and $C_t$ are the concentrations of MB at the start and at time $t$ (mg l$^{-1}$), respectively.

The concentration of Mn$^{2+}$ in the degradation was measured by atomic absorption spectroscopy (AAS; Persee TAS-990).

## 2.4. Identification of transformation products

The HPLC-MS system (Agilent 1200-6410B, USA) was equipped with an Agilent Poroshell EC-C18 column (100 mm × 4.6 mm packed with 2.7 μm particle size). The mobile phase consisted of 10 mM ammonium acetate (adjusted to pH 5.3 by acetic acid) and acetonitrile at a ratio of 78 : 22 (v/v), and the flow rate was 0.8 ml m$^{-1}$ with an injection volume of 20 μl. The HPLC separation was coupled with an ion trap mass spectrometer, equipped with an electrospray ionization (ESI) source and operated in positive mode. The ESI conditions were as follows: capillary voltage at 3.5 kV, endplate offset fixed at 500 V, skimmer at 40 V, trap drive at 53 V, a nebulizer pressure of 70 psi, a drying gas flow of 12 l min$^{-1}$ and a drying temperature of 350°C. The mass range was 50–700 $m/z$.

# 3. Results and discussion

## 3.1. Characterization of MnO$_x$

The crystalline structure of MnO$_x$ was characterized by XRD. According to the XRD patterns shown in figure 1*a*, the diffraction peaks (2$\theta$) of 12.74°, 37.63°, 49.90°, 56.18°, 60.24°, 65.52° and 73.07° correspond, respectively, to the (1 1 0), (3 1 0), (3 0 1), (4 1 1), (6 0 0), (5 2 1) and (4 5 1) crystal planes of α-MnO$_2$ (JCPDS card no. 72-1982) [39]. The diffraction peaks of 12.8°, 18.1°, 37.4°, 42°, 49.8° and 60.2°could be indexed to α-MnO$_2$ (JCPDS card no. 81-1947) [40]. The diffraction peaks of 37.12°, 42.66°, 56.02°, 66.76°, 75.02° and 78.92° match with the (1 0 0), (1 0 1), (1 0 2), (1 1 0), (1 0 3) and (2 0 0) crystal planes of ε-MnO$_2$ (JCPDS card no. 30-0820) [41]. The diffraction peaks of 13.45°, 18.20°, 18.59°, 21.06°, 21.32°, 37.43° and 38.28°could be indexed to todorokite (JCPDS card no. 84-1713) [42].

The sample of MnO$_x$ was also characterized using FTIR spectroscopy, which is a powerful tool to study the vibrational behaviour of lattices and provide crystalline phases of MnO$_x$ in the amorphous state. The FTIR spectra of MnO$_x$ are shown in figure 1*b*. MnO$_x$ has two strong bands located at 538.16 and 580.59 cm$^{-1}$, which are the characteristic adsorption peaks of MnO$_2$ (Mn–O), and two broad bands attributed to the Mn$^{3+}$–O bending–stretching vibration of todorokite could also be detected,

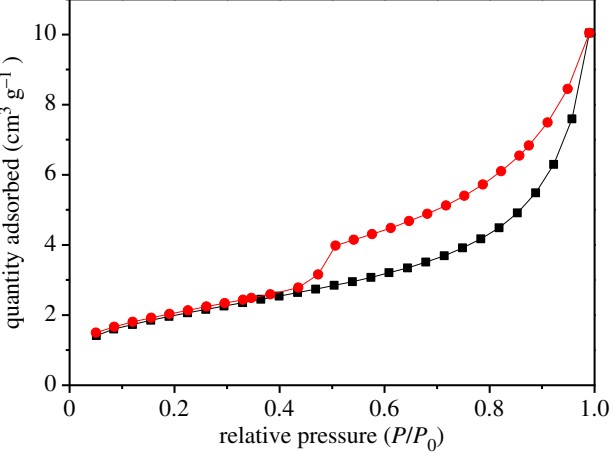

**Figure 2.** The N$_2$ adsorption–desorption isotherms of MnO$_x$.

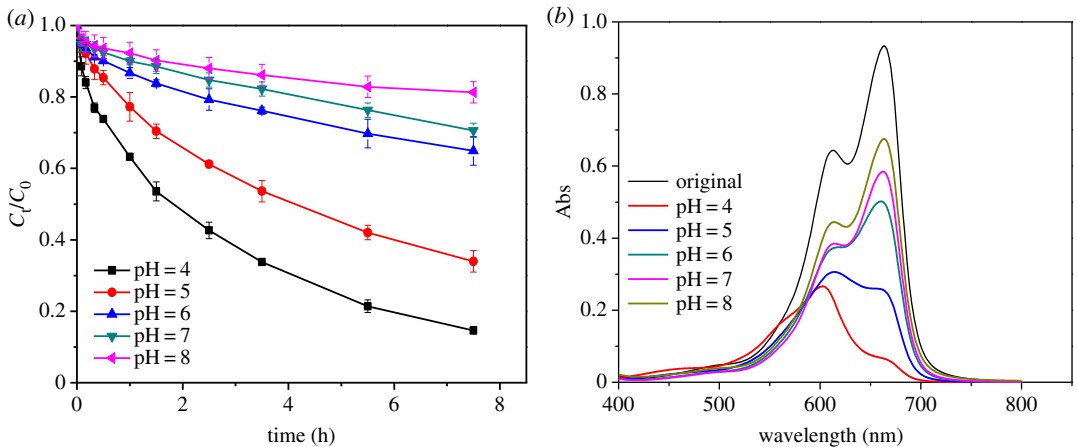

**Figure 3.** Degradation of MB over time in different pH (*a*) and photo-spectra of the reacted solution after twofold dilution (*b*) ([MB] = 10 mg l$^{-1}$, [MnO$_x$] = 1 g l$^{-1}$).

which were located at 468.72 and 1033.89 cm$^{-1}$ [43–46]. In addition, the vibration peak at 3291.66 cm$^{-1}$ was attributed to the symmetrical stretching vibration of hydroxyl (−OH) formed by physical adsorption of water molecules or ion exchange.

The N$_2$ sorption and desorption isotherms are shown in figure 2. According to the IUPAC classification, the N$_2$ adsorption isotherm closely resembles a type IIb isotherm [47] with a weak adsorption capacity. Within a low relative pressure ($P/P_0 < 0.3$), the adsorbed volume of N$_2$ increased slowly with relative pressure. Under the effect of monolayer adsorption, the adsorption and desorption lines almost coincided with each other, which was also confirmed by Tang *et al.* [48]. In addition, the adsorbed volume of N$_2$ rapidly increased under high relative pressure ($P/P_0 > 0.3$), which has been regarded as a character of capillary condensation (relative pressure ranging from 0.45 to 0.9) [49]. The specific surface area of MnO$_x$ was 7.17 m$^2$ g$^{-1}$.

In the electronic supplementary material, figure S1 depicts the surface morphology characteristics of MnO$_x$ by SEM. Particles with different shapes and size were spread in the field. Many grain structures were stuck to the surface of larger particles and some rod-like structures existed between the gaps of the particles. Such observations indicated that the surface morphology material is irregular.

## 3.2. Degradation of MB

### 3.2.1. pH effects

An experiment was conducted to investigate the relationship between the value of pH and the removal efficiency of MB by MnO$_x$, and the results are shown in figure 3*a*. The degradation was promoted by lower pH values and a significant gap existed between pH 5 and 6. The degradation of MB reached

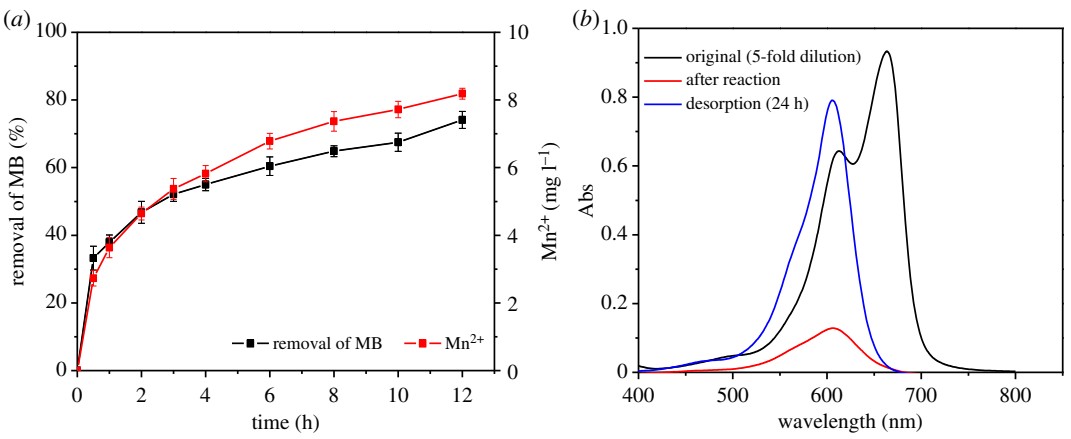

**Figure 4.** Concentration profile of $Mn^{2+}$ in the degradation of MB (*a*) and photo-spectra of the reaction and desorption solution (*b*) (pH = 4, $[MnO_x]$ = 3 g $l^{-1}$, $[MB]$ = 30 mg $l^{-1}$, reaction time = 10 h, desorption time = 24 h).

66.1% in 7.5 h at pH 5 and increased to 85.6% at pH 4. However, it was less than 50% with pH values of 6, 7 and 8, in good accordance with previous reports that $MnO_x$ is more active at low pH [40,49,50].

Firstly, the value of pH would affect the surface charge of $MnO_x$ and finally influences the removal behaviour of MB on $MnO_x$ [51]. The zero point charge ($pH_{zpc}$) of $MnO_x$ was measured and is shown in electronic supplementary material, figure S2. When the pH was lower than 6.13, the surface charge of the $MnO_x$ was positive owing to the protonation. Then the adsorption could be held back by the electrostatic repulsion between the cationic MB ($MB^+$) and the surface active sites of $MnO_x$. When the pH was higher than 6.13, the surface of $MnO_x$ was negatively charged owing to the de-protonation reaction, which led to the formation of precursors between $MB^+$ and $MnO_x$ by mutual attraction. However, the result showed that the removal of MB by $MnO_x$ under acidic conditions was much higher than that under alkaline conditions.

In addition, a small amount of MB adsorbed onto the surface of $MnO_x$ would subsequently be rapidly transferred by oxidation in the low-pH environment. As shown in figure 3*b*, the photo-spectra of the sample at pH 4 presented a blue shift after the reaction with a decrease in the absorbance peak at 664 nm and an emerging peak at 605 nm, which might belong to the intermediates from the degradation process [40,52]. On the contrary, the absorbance peak patterns did not change under alkaline conditions as the oxidability of $MnO_x$ would be restrained. A further proof came from the increased concentration of $Mn^{2+}$ with the removal of MB in the reaction solution (figure 4*a*) as a reduced product of $MnO_x$.

A desorption test was conducted with the reacted $MnO_x$ by adding an equal volume of methanol to the reaction solution. It was found that the adsorbed MB only accounted for 0.36% of the total removed MB. The desorption solution also showed a single absorbance peak at 605 nm (figure 4*b*). Therefore, it is highly possible that oxidation might play a major role in removing MB.

### 3.2.2. Dosage of $MnO_x$ and initial concentration of MB

To check the effect of $MnO_x$ dosage on the degradation efficiency, a varying dosage of $MnO_x$ from 1 to 5 g $l^{-1}$ was studied at pH 4 and MB concentration of 10 mg $l^{-1}$. As shown in figure 5*a*, with the increase in the amount of $MnO_x$, the degradation of MB significantly increased. It is worth noting that the degradation of MB reached 69% and 97% in 5 min and 60 min, respectively, under the dosage of 5 g $l^{-1}$ of $MnO_x$. That might be due to the availability of the reactive sites in the system [49].

The initial concentration of MB was also investigated with a range from 10 to 50 mg $l^{-1}$ at pH 4 with an $MnO_x$ dosage of 3 g $l^{-1}$ as presented in figure 5*b*. It can be clearly seen that the degradation percentage increases with time, and also decreases with increasing initial concentration. Over 99% of MB was degraded in 5.5 h with an initial concentration of 10 mg $l^{-1}$, much higher than that with 50 mg $l^{-1}$ MB (52%), which might be due to the limited amount of active sites of $MnO_x$. Moreover, the intermediates formed during the degradation of MB might compete with the MB molecules for the available active sites.

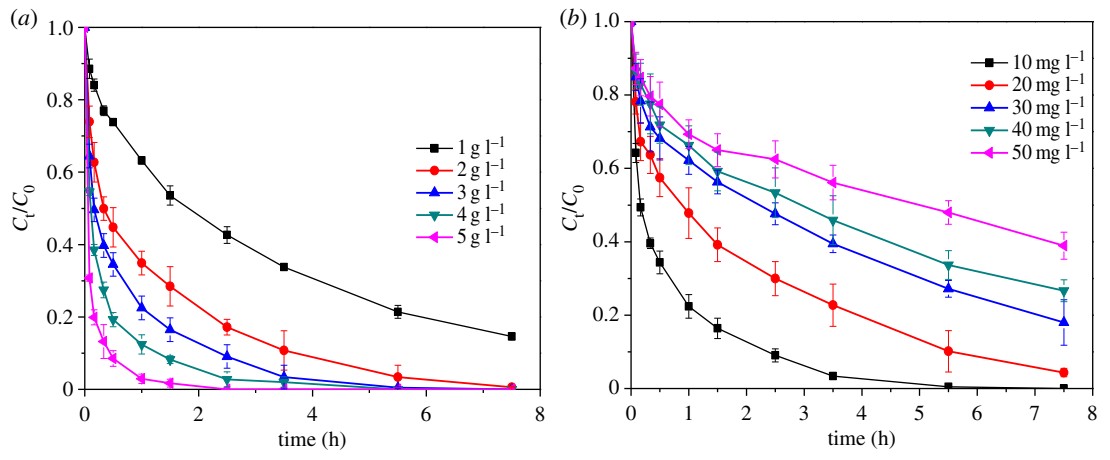

**Figure 5.** Degradation of MB with the various dosages of MnO$_x$ (*a*) and initial concentrations of MB (*b*).

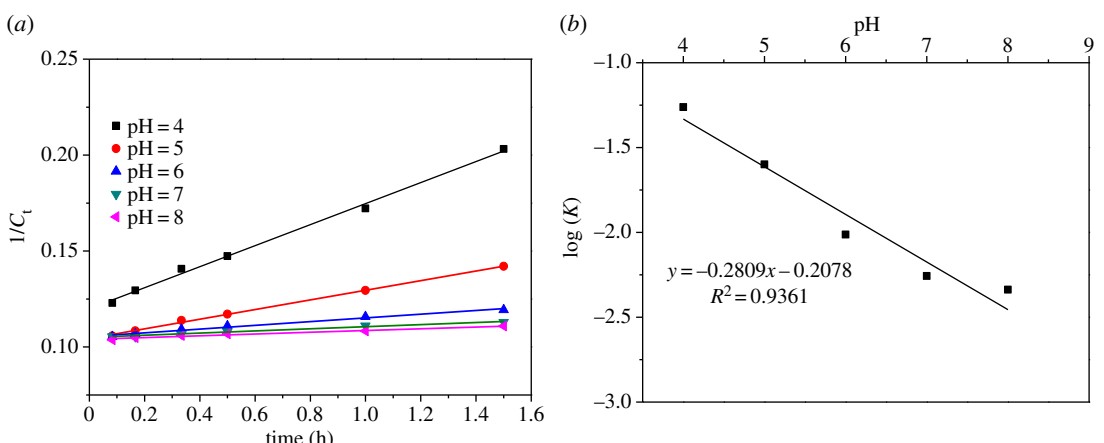

**Figure 6.** Second-order kinetic plots at different pH values (*a*) and the correlation between kinetic constants and pH values (*b*).

## 3.3. Kinetics analysis

The surface binding model of MnO$_2$ with organic compounds, proposed by Arimi *et al.* [35] and Stone & Morgan [53], can be used to describe the reaction process,

$$\vdots MnOH + MB \overset{k_1,k_2}{\leftrightarrow} \vdots MnMB + H_2O, \tag{3.1}$$

$$\vdots MnMB \overset{k_3}{\to} Mn^{2+} + products \tag{3.2}$$

and

$$[total\ sites] = [\vdots MnOH] + [\vdots MnMB], \tag{3.3}$$

where $\vdots$MnOH is the number of free active sites on the surface of NMO; MB is the concentration in the aqueous solution; and $\vdots$MnMB, a manganese–methylene blue complex, is formed on the surface of MnO$_x$ and releases Mn$^{2+}$ into the solution.

The dissolving procedure of the products in the solution can be described by equation (3.2). Therefore, the total active sites in the reaction system can be demonstrated in equation (3.3).

The second-order kinetic reaction equation is used to simulate the obtained data as follows:

$$\frac{d[MB]}{dt} = -k \cdot [MB]^2. \tag{3.4}$$

In the above equation, [MB] is the concentration of MB at time $t$. The second-order kinetic reaction constant $k$ can be obtained by a straight line by fitting 1/[MB] versus $t$, and the plots of the experimental results are shown in figure 6. With the above model, figure 6*a* shows the fitting plots. The calculated kinetic data are summarized in table 1 and the kinetic constant $k$ of pH = 4 was far higher than that of the other four pH values, as discussed above.

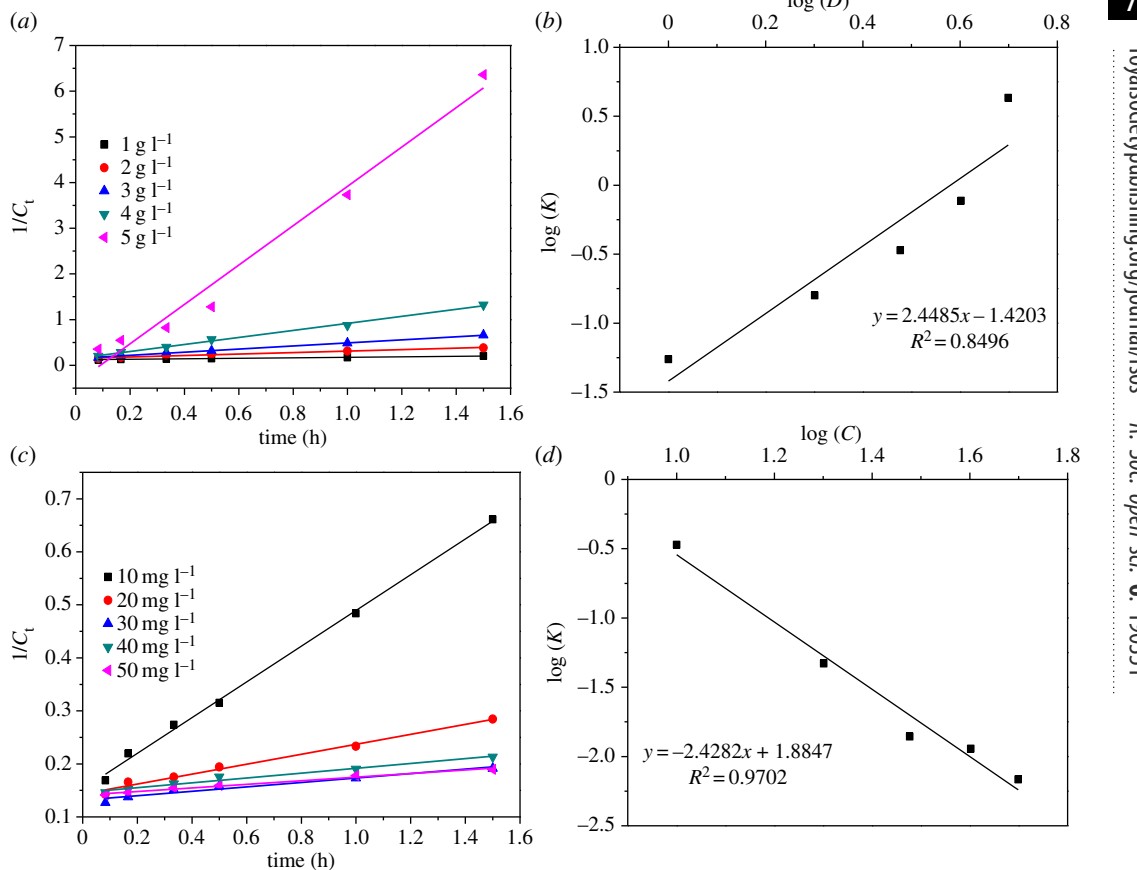

**Figure 7.** Second-order kinetics plots at varying dosages of $MnO_x$ (*a*) and initial concentrations of MB (*c*) and the correlation between the obtained kinetic constants and the dosages of $MnO_x$ (*b*) and the initial concentrations of MB (*d*).

Previous studies have already demonstrated that $H^+$ in solution may promote the redox capacity of $MnO_2/Mn^{2+}$ and a mathematical correlation exists between the kinetic constant and the pH [34,35]:

$$MnO_2 + 4H^+ + 2e^- \leftrightarrow Mn^{2+} + 2H_2O \qquad E_H^0 = 1.29\,V\,at\ \ 25°C \tag{3.5}$$

and

$$K' = K_{pH} \cdot [H^+]^m, \text{or } \log K' = \log K_{pH} + m \cdot [H^+]. \tag{3.6}$$

Here, $K'$ is a constant independent of the solution pH, $K_{pH}$ represents the constant obtained from the fitted kinetics and $m$ is a constant standing for the reaction series. As shown in figure 6*b*, the results fit well with equation (3.6), and the values of $K'$ and $m$ are 0.2078 and 0.2809, respectively. The value of $m$ is higher than the data in previous studies on the treatment of organic material by manganese dioxide [35].

Figure 7*a,c* shows the fitting plots of the second-order kinetics at various dosages of $MnO_x$ and at different initial concentrations of MB. The kinetic constants can be seen in table 1. In addition, a double logarithmic correlation was also found between the obtained kinetic constants and the dosages of $MnO_x$ or the initial concentrations of MB, as shown in figure 7*b,d*, respectively.

## 3.4. Proposed intermediates and reaction pathway

The photo-spectrum of the reaction solution indicated a highly possible chemical transformation of MB with $MnO_x$, as shown in figure 3*b*. To gain further understanding of the reaction, the sample was analysed with HPLC-MS to identify the intermediates formed after a 7.5 h reaction. The mass spectra are shown in figure 8, where several $m/z$ peaks can be clearly noted, representing the formed intermediates. Some peaks ($m/z$ 256.1, 270.1) have also been found in other studies after the oxidation of MB [4,49].

**Table 1.** Parameters of second-order kinetic equations with different influencing factors.

| pH | | | | dosage of MnO$_x$ | | | | initial concentration | | | |
|----|---|---|---|---|---|---|---|---|---|---|---|
| pH | $K$ | $R^2$ | s.d. | g l$^{-1}$ | $K$ | $R^2$ | s.d. | mg l$^{-1}$ | $K$ | $R^2$ | s.d. |
| 4 | 0.0547 | 0.9953 | 0.0026 | 1 | 0.0547 | 0.9953 | 0.0026 | 10 | 0.3371 | 0.9970 | 0.0264 |
| 5 | 0.0252 | 0.9979 | 0.0006 | 2 | 0.1590 | 0.9770 | 0.0217 | 20 | 0.0469 | 0.9861 | 0.0108 |
| 6 | 0.0097 | 0.9786 | 0.0010 | 3 | 0.3371 | 0.9970 | 0.0303 | 30 | 0.0140 | 0.9379 | 0.0046 |
| 7 | 0.0055 | 0.9811 | 0.0004 | 4 | 0.7723 | 0.9952 | 0.0424 | 40 | 0.0114 | 0.9723 | 0.0044 |
| 8 | 0.0046 | 0.9624 | 0.0005 | 5 | 4.3070 | 0.9736 | 0.3644 | 50 | 0.0068 | 0.9802 | 0.0023 |

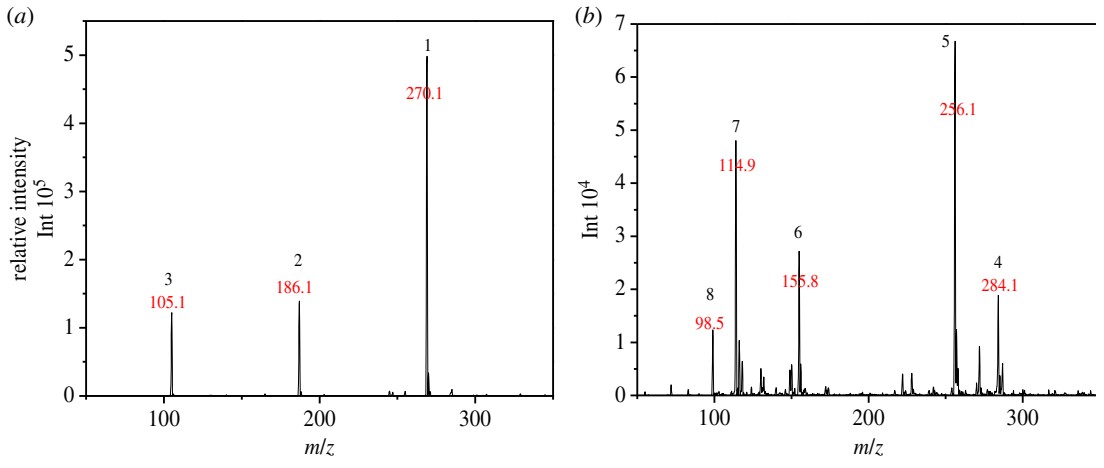

**Figure 8.** (*a,b*) Mass spectrum of MB intermediates in eluent after a 7.5 h reaction.

**Table 2.** Suggested intermediate structures according to analysis by HPLC-MS.

| no. | m/z | suggested structure |
| --- | --- | --- |
| 1 | 270.1 | |
| 2 | 186.1 | |
| 3 | 105.1 | |
| 4 | 284.1 | |
| 5 | 256.1 | |
| 6 | 155.8 | |
| 7 | 114.9 | |
| 8 | 98.5 | |

As shown in figure 8*a*, some fragments were observed in eluent at 2.12 min by mass spectra. Those peaks are the possible reduced products in the oxidation of MB by $MnO_x$, which are eluted at the same time as one of the major products, azure A (AA). Similarly, other fragments are eluted with another major product, azure B (AB), at 3.14 min (figure 8*b*). But the content of these small fragments is very low relative to the other four main intermediates (azure A, azure B, azure C, thionin). Accordingly, these possible intermediates are listed in table 2.

In addition, the detected intermediates are in good accordance with the theory of molecular bond energy. In a molecule of MB, the bonds of $N–CH_3$ connected to $C^7$ or $C^{12}$ have the lowest bond energy (electronic supplementary material, table S1) and thus would be broken first in the oxidation, resulting in demethylation, such as mono-demethylation, di-demethylation, tri-demethylation and

**Figure 9.** The probable demethylation of MB molecules by MnO$_x$.

complete demethylation of nitrogen. Ring opening might subsequently occur. Similar results were also observed previously [4,54,55]. A possible demethylation pathway of MB is shown in figure 9.

## 4. Conclusion

Compared with other synthetic oxidants, Fenton or ozone technology, the current study demonstrates that this kind of natural manganese oxide could be a potentially efficient low-cost oxidant for the degradation of MB in aqueous solution. The results show that the reaction preferred an acidic pH and was strongly influenced by the dosage of MnO$_x$ and the initial concentration of MB. The second-order kinetic model can describe the reaction very well and a double logarithmic correlation fits well between the kinetic constants and the dosages of MnO$_x$ or the initial concentrations of MB. Several intermediates have been detected, which, linking to the theory of molecular bond energy, demonstrates that the degradation of MB is initiated by the demethylation.

Data accessibility. Additional data are available in the electronic supplementary material.

Authors' contributions. S.Z. carried out the degradation experiment, analysed the data and drafted the manuscript; Z.D. and X.L. carried out the analysis of the dynamic experiment data; Yu.Z. and Y.H. conducted the instrumental testing and analysis and revised the manuscript; Yo.Z. conceived of the study, designed the research, coordinated the study and helped revise the manuscript.

Competing interests. We declare we have no competing interests.
Funding. The study is funded by the Natural Science Research Program of Jiangsu Province for Colleges and Universities (grant no. 18KJA610001, 18KJB610008) and the Natural Science Foundation of Jiangsu Province (no. BK20160989).
Acknowledgements. The authors appreciate the assistance of Mr Wei Wang from the State Key Laboratory of Plateau Ecology and Agriculture, China.

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
