## [Reviewer comments · Royal Society Open Science]

Review History

RSOS-190351.R0 (Original submission)

Review form: Reviewer 1

Is the manuscript scientifically sound in its present form?

Yes

Are the interpretations and conclusions justified by the results?

Yes

Is the language acceptable?

Yes

Is it clear how to access all supporting data?

Yes

Do you have any ethical concerns with this paper?

No

Have you any concerns about statistical analyses in this paper?

No

Recommendation?

Accept with minor revision (please list in comments)

Comments to the Author(s)

- 1- Indicate the purity of each chemical reagent used
- 2- Characterize MnO_x by FTIR spectroscopy
- 3- Indicate the structure of crystalline MnO_x based on XRD and FTIR
- 4- Fit the experimental data to adsorption isotherms such as Langmuir and Freundlich
- 5- It would be interesting to study the effect of temperature on the adsorption process
- 6- If possible, propose a mechanism for MB degradation by MnO_x

Review form: Reviewer 2**Is the manuscript scientifically sound in its present form?**

Yes

Are the interpretations and conclusions justified by the results?

Yes

Is the language acceptable?

Yes

Is it clear how to access all supporting data?

Yes

Do you have any ethical concerns with this paper?

No

Have you any concerns about statistical analyses in this paper?

No

Recommendation?

Major revision is needed (please make suggestions in comments)

Comments to the Author(s)

Degradation of methylene blue by natural manganese oxides: Kinetics and transformation products

The article is reported the degradation of MB by natural manganese oxides and its transformation. Articles is needed much more modifications to achieve its standard.

Comments

1. Abstract is too shallow. It should be re-written.
2. Introduction is too short. Content is lag for the provided number of references.
3. Novelty is missing.
4. Manuscript seems like a report rather than a scientific article.
5. Scientific discussions are missing throughout the manuscript.
6. Confirmation of MnO_x ore is not enough.
7. Application part is only emphasized in the manuscript.

8. Significance of the material and its preliminary strong evidence for confirmation is lagging thoroughly.
9. Conclusion is also being as lack of scientific summary of the entire manuscript.
10. Why does the reaction need low pH values?
11. Comparison of the results with earlier literature will make the article sensible.
12. No new information could obtain from this manuscript.
13. Reference style is very random and non-uniform. Make it in uniform order corresponds to the journal reference style.

Decision letter (RSOS-190351.R0)

10-May-2019

Dear Professor Zhang:

Title: Degradation of methylene blue by natural manganese oxides: Kinetics and transformation products

Manuscript ID: RSOS-190351

The editor assigned to your manuscript has now received comments from reviewers. I apologise that this took longer than usual.

We would like you to revise your paper in accordance with the referee and Subject Editor suggestions which can be found below (not including confidential reports to the Editor). Please note this decision does not guarantee eventual acceptance.

Please submit your revised paper before 02-Jun-2019. Please note that the revision deadline will expire at 00.00am on this date. If we do not hear from you within this time then it will be assumed that the paper has been withdrawn. In exceptional circumstances, extensions may be possible if agreed with the Editorial Office in advance. We do not allow multiple rounds of revision so we urge you to make every effort to fully address all of the comments at this stage. If deemed necessary by the Editors, your manuscript will be sent back to one or more of the original reviewers for assessment. If the original reviewers are not available we may invite new reviewers.

Please also include the following statements alongside the other end statements. As we cannot

publish your manuscript without these end statements included, if you feel that a given heading is not relevant to your paper, please nevertheless include the heading and explicitly state that it is not relevant to your work.

- Ethics statement

Please clarify whether you received ethical approval from a local ethics committee to carry out your study. If so please include details of this, including the name of the committee that gave consent in a Research Ethics section after your main text. Please also clarify whether you received informed consent for the participants to participate in the study and state this in your Research Ethics section.

OR

Please clarify whether you obtained the necessary licences and approvals from your institutional animal ethics committee before conducting your research. Please provide details of these licences and approvals in an Animal Ethics section after your main text.

OR

Please clarify whether you obtained the appropriate permissions and licences to conduct the fieldwork detailed in your study. Please provide details of these in your methods section.

On behalf of the Subject Editor Professor Anthony Stace and the Associate Editor Professor Tobias Hertel.

RSC Associate Editor:
Comments to the Author:
(There are no comments.)

RSC Subject Editor:
Comments to the Author:
(There are no comments.)

Reviewers' Comments to Author:
Reviewer: 1

Comments to the Author(s)
1- Indicate the purity of each chemical reagent used
2- Characterize MnOx by FTIR spectroscopy

- 3- Indicate the structure of crystalline MnO₂ based on XRD and FTIR
- 4- Fit the experimental data to adsorption isotherms such as Langmuir and Freundlich
- 5- It would be interesting to study the effect of temperature on the adsorption process
- 6- If possible, propose a mechanism for MB degradation by MnO₂

Reviewer: 2

Comments to the Author(s)

Degradation of methylene blue by natural manganese oxides: Kinetics and transformation products

The article is reported the degradation of MB by natural manganese oxides and its transformation. Articles is needed much more modifications to achieve its standard.

Comments

1. Abstract is too shallow. It should be re-written.
2. Introduction is too short. Content is lag for the provided number of references.
3. Novelty is missing.
4. Manuscript seems like a report rather than a scientific article.
5. Scientific discussions are missing throughout the manuscript.
6. Confirmation of MnO₂ ore is not enough.
7. Application part is only emphasized in the manuscript.
8. Significance of the material and its preliminary strong evidence for confirmation is lagging thoroughly.
9. Conclusion is also being as lack of scientific summary of the entire manuscript.
10. Why does the reaction need low pH values?
11. Comparison of the results with earlier literature will make the article sensible.
12. No new information could obtain from this manuscript.
13. Reference style is very random and non-uniform. Make it in uniform order corresponds to the journal reference style.

Author's Response to Decision Letter for (RSOS-190351.R0)

See Appendix A.

Decision letter (RSOS-190351.R1)

12-Jun-2019

Dear Professor Zhang:

Title: Degradation of methylene blue by natural manganese oxides: Kinetics and transformation products

Manuscript ID: RSOS-190351.R1

It is a pleasure to accept your manuscript in its current form for publication in Royal Society Open Science. The chemistry content of Royal Society Open Science is published in collaboration with the Royal Society of Chemistry.

On behalf of the Subject Editor Professor Anthony Stace and the Associate Editor Professor Tobias Hertel.

RSC Associate Editor
Comments to the Author:
(There are no comments.)

Reviewer(s)' Comments to Author:

Appendix A

Response to reviewer comments

(Manuscript ID: RSOS-190351)

We highly appreciate the valuable comments of reviewers that are helpful for significantly improving the manuscript quality. The manuscript has been thoroughly revised accordingly. Each points of the comments are responded as follows.

Reviewer #1:

1. Indicate the purity of each chemical reagent used

Revised.

After revision:

Page 2 in the part of materials and chemical agents: Methylene blue (98.5%), acetic acid (99.8%), sodium acetate (98%), sodium hydrogen phosphate (99%), hydrochloric acid (36-38%), and sodium hydroxide (96%) are not further purified and are purchased from Sinopharm Chemical Reagent Co., Ltd (Shanghai). Natural manganese ore (MnOx, with a relative content of MnO₂ 75.38%)...

2. Characterize MnOx by FTIR spectroscopy

We have added the FTIR spectroscopy and analyzed the crystalline of MnOx.

After revision:

Page 4 in the characterization of MnOx section: The sample of MnOx was also characterized using FTIR spectroscopy, which was the powerful tool to study vibrational behavior of lattices and provide crystalline phases of MnOx in amorphous. The FTIR spectra of MnOx were shown in figure 1b. MnOx has two strong band located at 538.16 cm⁻¹ and 580.59 cm⁻¹, which were the characteristic adsorption peaks of MnO₂ (Mn-O) and the presence of two broad bands attributed to Mn³⁺-O bending stretching vibration of todorokite could also be detected, which were located at 468.72cm⁻¹ and 1033.89 cm⁻¹ [43-46]. In addition, the vibration peak at 3291.66 cm⁻¹ was attributed to the symmetrical stretching vibration of hydroxyl (-OH) formed by physical adsorption of water molecules or ion exchange.

3. Indicate the structure of crystalline MnOx based on XRD and FTIR

The article presented a natural manganese ore whose composition is more complex and consists of manganese oxides, manganese hydroxides and some natural organic compounds, as shown in XRD patterns. The FTIR was added to further reveal the structure of crystalline MnOx.

The result was followed by comment 2.

4. Fit the experimental data to adsorption isotherms such as Langmuir and Freundlich

Surely the removing process of MB must start with adsorption onto MnOx. However, the adsorbed MB should be subsequently oxidized, as demonstrated in our study with the shifted UV spectrum, the desorption test, and the identified intermediates. Therefore, the adsorption isotherms were not applied.

5. It would be interesting to study the effect of temperature on the adsorption process

The temperature may give an effect in the degradation of MB. But, the work was focus on exploring the reaction style and the degradation pathway in room temperature. What's more, the experiment data has proved that the removal of MB was due to the chemical oxidation degradation by MnOx in low pH.

6. If possible, propose a mechanism for MB degradation by MnOx

The degradation mechanism for MB by MnOx has been suggested in the section 4 of the manuscript. We concluded that the degradation of MB was initiated with the demethylation, and the detection results of the intermediates also have verified the hypothesis.

Reviewer #2:

1. Abstract is too shallow. It should be re-written.

We have revised the abstract.

After revision:

Page 1 in the abstract section. Though the surface area of MnOx was only 7.17 m²/g, it performed well in the degradation of MB with a removal percentage 85.6% in pH 4. Furthermore, several transformation products of MB were identified with HPLC-MS, which was linked with the bond energy theory to reveal that the degradation was initiated with demethylation.

2. Introduction is too short. Content is lag for the provided number of references.

Many new references were cited in the introduction section.

After revision:

Page 2 in the introduction part. New articles (14, 15, 17, 18, 29, 30, 37) were cited to better support the introduction and other articles (40, 52) were cited to discuss.

3. Novelty is missing.

The purpose of the study was to supply a potential environmental material with cheap price, easy to operation and from nature to treat wastewater. The research has revealed an environment friendly material (MnOx) used to degraded the dyestuffs wastewater with a huge reserves in nature.

4. Manuscript seems like a report rather than a scientific article.

The research has reported an available environmental material for treating dyestuffs wastewater and proved the active style between the MnOx and MB in low pH. The single factor experiments (pH, the dosage of MnOx and the concentration of MB) were also conducted to study their effects on the removal of MB and the dynamic process. On the other hand, the probable degradation course was supposed followed the checked intermediates and the bond energy theory, from which the study concluded that the degradation was initiated with demethylation.

5. Scientific discussions are missing throughout the manuscript.

In the discussion part of the article, we firstly make the reaction form clear that the chemical oxidation degradation was stimulated in low pH, which was proved as following: (1) the full wavelength scanning of the samples; (2) the concentration of Mn^{2+} in the sample; (3) the desorption experiment was conducted. Then the effects on the degradation of dosage of MnOx and the concentration of MB were also discussed, respectively. The single factor experiments were performed to reveal the dynamic process which would be benefit for its application.

6. Confirmation of MnOx ore is not enough.

We have added the FTIR spectroscopy of MnOx to further confirm the crystalline of MnOx.

After revision:

Page 4 in the characterization of MnOx section: The sample of MnOx was also characterized using FTIR spectroscopy, which was the powerful tool to study vibrational behavior of lattices and provide crystalline phases of MnOx in amorphous. The FTIR spectra of MnOx were shown in figure 1b. MnOx has two strong band located at 538.16 cm^{-1} and 580.59 cm^{-1} , which were the characteristic adsorption peaks of MnO_2 (Mn-O) and the presence of two broad bands attributed to Mn^{3+} -O bending stretching vibration of todorokite could also be detected, which were located at 468.72 cm^{-1} and 1033.89 cm^{-1} [43-46]. In addition, the vibration peak at 3291.66 cm^{-1} was attributed to the symmetrical stretching vibration of hydroxyl (-OH) formed by physical adsorption of water molecules or ion exchange.

7. Application part is only emphasized in the manuscript.

In the practical application, MnOx has been extensively used in drinking water treatment for the removal of iron and manganese. The degradation mechanism and pathway of organic contaminants was seldom discussed, which would be the theoretical basis for its application. And the next step was to explore that MnOx

degrade municipal wastewater under the research.

8. Significance of the material and its preliminary strong evidence for confirmation is lagging thoroughly.

Beside XRD, we added FTIR into the manuscript to provide more info about the material. The result of FTIR was followed by comment 6.

9. Conclusion is also being as lack of scientific summary of the entire manuscript.

The work was designed to investigate the oxidation property of MnO_x for environmental containments and its influenced factors. The conclusion was supported by the following:

- (1) the study of the full wavelength scanning found new absorption peaks, and the characteristic absorption peaks of MB do not decrease proportionally;
- (2) the concentration of Mn²⁺ in the sample was simultaneous increasing with MB removal rate;
- (3) the contribution of adsorption was only 0.36% of the total removal percentage 90.5% in low pH value;
- (4) the results of LC-MS give a strong support to the chemical degradation of MB by MnO_x, and those intermediates were also checked.

10. Why does the reaction need low pH values?

For the Nernst equation $\text{MnO}_2 + \text{H}^+ = \text{Mn}^{2+} + \text{H}_2\text{O}$, increasing the concentration of H⁺ would promote the reaction, which means that increasing the concentration of H⁺ would raise the oxidability of MnO₂ and to degrade the contaminant. However, the alkaline condition would restrain the redox reaction, which would be not conducive to the containment removal.

11. Comparison of the results with earlier literature will make the article sensible.

The results of the article have been further revised under the earlier literatures.

After revision:

Page 5 in discussion part. Firstly, the value of pH would affect the surface charge of MnOx and finally influences the removal behavior of MB on MnOx [51]. The zero point charge (pHzpc) of MnOx was measured and shown in Fig. S2. When pH was lower than 6.13, the surface charge of the MnOx was positive due to the protonation. Then the adsorption could be held back by the electrostatic repulsion between the cationic methylene blue (MB⁺) and the surface active sites of MnOx. When pH was higher than 6.13, the surface of MnOx was negatively charged due to the de-protonation reaction, which would lead to the formation of precursors between MB⁺ and MnOx by a mutual attraction. However, the result showed that the removal of MB by MnOx in acidic condition was much higher than that of alkaline condition.

In addition, a small amount of MB adsorbed to the surface of MnOx would subsequently be rapidly transferred by oxidizing at low pH environment. As shown in Fig 3b, the photo-spectra of the sample at pH 4 presented a blue shift after the reaction with a decrement of the absorbance peak at 664 nm and an emerging peak at 605nm, which might belong to the intermediates from the degradation process [40, 52]. On the contrary, the absorbance peak patterns have not changed under alkaline conditions for it would restrain the oxidizability of MnOx. A further proof came from the increased concentration of Mn²⁺ with the removing of MB in reaction solution (Fig. 4a) as a reduced product of MnOx.

Page 7 in conclusion part. Compared to other synthetic oxidants, Feton or ozone technology, the current study demonstrates that this kind of natural manganese oxide could be a potentially efficient low-cost oxidant for the degradation of methylene blue in aqueous solution.

12. No new information could obtain from this manuscript.

The element Mn is an important participant involved in the earth biochemical cycle, which is the second largest metal element after iron and possess high redox potential. Abundant reserves of manganese ore have been surveyed on the earth's crust and ocean. The manuscript reported have proved the degradability of natural manganese

ore for MB, and also revealed the mechanisms of MnOx degrading MB, which would be theoretical significance for its application.

13. Reference style is very random and non-uniform. Make it in uniform order corresponds to the journal reference style.

We have unified the format of reference.

After revision

Page 9 in the part of references.